# Mitochondrial DNA copy number, metabolic syndrome, and insulin sensitivity: Insights from the Sugar, Hypertension, and Physical Exercise studies

**Stephanie Y. Yang**[1], **Caleb S. Mirabal**[1], **Charles E. Newcomb**[1], **Kerry J. Stewart**[2], **Dan E. Arking**[1]*

1 Department of Genetic Medicine, McKusick-Nathans Institute, Johns Hopkins University School of Medicine, Baltimore, Maryland, United States of America, 2 Department of Cardiology, Johns Hopkins University School of Medicine, Baltimore, Maryland, United States of America

* arking@jhmi.edu

## Abstract

Mitochondrial DNA copy number (mtDNA-CN) measured in blood has been associated with many aging-related diseases, with higher mtDNA-CN typically associated with lower disease risk. Exercise training is an excellent preventative tool against aging-related disorders and has been shown to increase mitochondrial function in muscle. Using the Sugar, Hypertension, and Physical Exercise cohorts (N = 105), we evaluated the effect of 6-months of exercise intervention on mtDNA-CN measured in blood. Although there was no significant relationship between exercise intervention and mtDNA-CN change (P = 0.29), there was a nominally significant association between mtDNA-CN and metabolic syndrome (P = 0.04), which has been seen in previous literature. We also identified a nominally significant association between higher mtDNA-CN and higher insulin sensitivity (P = 0.02).

## Introduction

Mitochondria are well-known for their essential roles in ATP production, though they perform additional functions such as calcium homeostasis, apoptosis signaling, and lipid metabolism [1–3]. ATP synthesis and supply is crucial for skeletal muscle contraction during exercise, and thus, functional mitochondria are necessary for aerobic exercise. Mitochondria contain their own genomes (mtDNA), which can range from tens to thousands of copies per cell. This variation in quantity is referred to as mitochondrial DNA copy-number (mtDNA-CN), and widely differs across cell types and individuals. Higher mtDNA-CN levels are positively associated with mitochondrial membrane potential, respiratory enzyme function, and energy reserves [4, 5], suggesting that mtDNA-CN may be a marker of mitochondrial health. Lower mtDNA-CN in buffy coat, the fraction of blood that contains leukocytes and platelets, has been associated with frailty, often characterized by decreased muscle tone [6]. Exercise has also been shown to be an excellent preventive tool for many of the aging-related disorders associated with lower mtDNA-CN [7, 8]. These findings suggest a relationship between exercise

**Data Availability Statement:** The minimal dataset necessary to replicate our analyses is available on

Figshare: https://figshare.com/articles/dataset/shape_for_release_txt/20152202.

**Funding:** This work was supported by NHLBI grants R01HL13573 (SYY, CSM, CEN, DEA) and R01HL144569 (SYY, CSM, CEN, DEA). SHAPE projects were supported by NIDDK and NICRR grants R01DK062368 (KJS) and UL1RR025005 (KJS). The funders had no role in study design, data collection and analysis, decision to publish, or preparation of the manuscript.

**Competing interests:** The authors have declared that no competing interests exist.

training, which can increase muscle density [9] and mtDNA-CN measured in blood. Indeed, persistent exercise training has been shown to increase mitochondrial function and mitochondrial volume in skeletal muscle [10, 11]. Additionally, Lanza et. al has shown that mtDNA-CN in the muscle of endurance-trained individuals is higher than that of sedentary subjects [8]. However, mtDNA-CN in skeletal muscle is difficult to obtain, as muscle biopsy is required. We hypothesize that exercise intervention can increase mtDNA-CN in blood. To answer this question, we used two randomized controlled exercise intervention cohorts. Participants were aged 30–65 years, and performed aerobic and resistance exercise training 3 times a week for a duration of 6 months.

## Results

### Exercise increases VO$_2$max and decreases BMI in the SHAPE cohorts

The Sugar, Hypertension, and Physical Exercise (SHAPE) cohorts are a set of randomized controlled studies that aimed to evaluate the effects of exercise and diet interventions on blood pressure and other secondary outcomes [12]. Briefly, participants were randomized into two intervention groups, which varied based on the specific SHAPE study (Table 1). Interventions were for 6 months, and blood samples were drawn at both baseline (pre-intervention) and final (post-intervention) visits.

Maximal oxygen uptake (VO$_2$max), is a measurement of an individual's aerobic capacity and increases after exercise training [13, 14]. To confirm the efficacy of the exercise intervention, we examined associations between the 6-month change in VO$_2$max and the number of exercise sessions that an individual attended. There was a positive association (R = 0.38, P = $7.44 \times 10^{-5}$) between more exercise sessions and a 6-month increase VO$_2$max (S1 Fig). Additionally, the number of exercise sessions was significantly associated with a 6-month decrease in BMI (R = -0.29, P = 0.002, S2 Fig). Taken together, these correlations indicate that exercise intervention was effective.

### Measurement and validation of mtDNA-CN in the SHAPE cohorts

mtDNA-CN was measured from whole blood samples obtained at baseline and final visits. Briefly, a monochrome qPCR assay with a nuclear target (albumin) and a mitochondrial target (D-loop) was used to measure the proportion of mitochondrial DNA relative to nuclear DNA [15]. To avoid batch effects, samples derived from the same individual were run on the same plate, and the final mtDNA-CN metric was adjusted for plate as a random effect. The final data was then centered and scaled.

mtDNA-CN is typically higher in females and decreases with age [16, 17]. To validate our mtDNA-CN metric, we evaluated associations with these two known covariates, using only baseline (pre-intervention) samples (N = 145). Despite the small sample size, age and sex were both significantly associated in the expected directions [18] (Fig 1). As it has been shown that the relationship between mtDNA-CN and age is nonlinear [19], We modeled the effect of age on mtDNA-CN using a natural spline, yielding a knot at 52.6 years. However, using a log

**Table 1. SHAPE cohort data.**

| Study | N | Comorbidities | Group1 | Group2 | Completed Protocol |
|-------|---|---------------|--------|--------|--------------------|
| SHAPE3 | 77 | Overweight/obese + prediabetes/diabetes | Diet | Diet + Exercise | 55 |
| SHAPE5 | 77 | Obese, otherwise healthy | Exercise + Low CHO | Exercise + Low Fat | 60 |

Cohort data for the SHAPE3 and SHAPE5 cohorts. Low CHO = low carbohydrate weight loss diet, Low Fat = low fat weight loss diet.

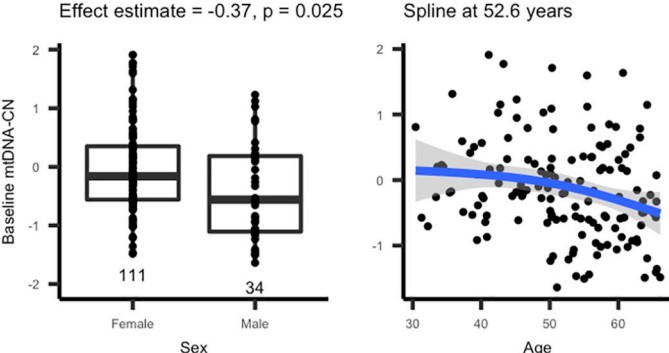

**Fig 1. Associations between baseline mtDNA-CN and known covariates.** Baseline mtDNA-CN is associated with age and sex in the expected directions. Females have higher baseline mtDNA-CN, and baseline mtDNA-CN decreases with age.

likelihood test, the spline age term did not perform significantly better than a linear term (P = 0.42), potentially due to the small sample size. Adding sex as a covariate significantly improved the model (P = 0.008).

These effect estimates are consistent across the two different SHAPE studies, confirming that the associations are not driven by any one study (S3 Fig).

During quality control checks, we discovered that higher baseline mtDNA-CN was significantly associated with study dropout (R = 0.21, P = 0.01, S4 Fig). This association persisted even when stratifying the analysis by SHAPE study (S5 Fig). To understand what could be driving the relationship between baseline mtDNA-CN and study completion, we examined associations between study completion and several other variables. However, none of these potential explanatory variables was significantly associated with study completion. We note that increased age was nominally significantly associated with dropout, however, since increased age is associated with lower mtDNA-CN, this would not explain the observed relationship between higher mtDNA-CN and increased rates of study dropout (S1 Table). After these analyses, we were unable to account for the variation in dropout explained by baseline mtDNA-CN, and currently do not have a biological explanation for this finding.

## mtDNA-CN is correlated between visits

Because baseline and final measurements are only separated by six months, we expected baseline and final mtDNA-CN to be correlated. After correcting for plate effects, the Pearson correlation was 0.578 (N = 105) and is consistent between the two studies (Fig 2).

## No significant change in mtDNA-CN after 6 months of study intervention

To calculate the change in mtDNA-CN, we subtracted baseline mtDNA-CN from final mtDNA-CN. As such, positive values indicate an increase in mtDNA-CN over the 6-month period. We found that more extreme baseline mtDNA-CN measurements were likely to have larger 6-month changes, suggesting a reversion to the mean (S6 Fig). To account for this, all analyses evaluating associations with 6-month change in mtDNA-CN are adjusted for baseline mtDNA-CN as a covariate.

When comparing the change in mtDNA-CN between exercisers and non-exercisers, there was no significant difference in the mtDNA-CN change (P = 0.29, Fig 3). We also analyzed associations between the number of exercise sessions attended and change in mtDNA-CN in

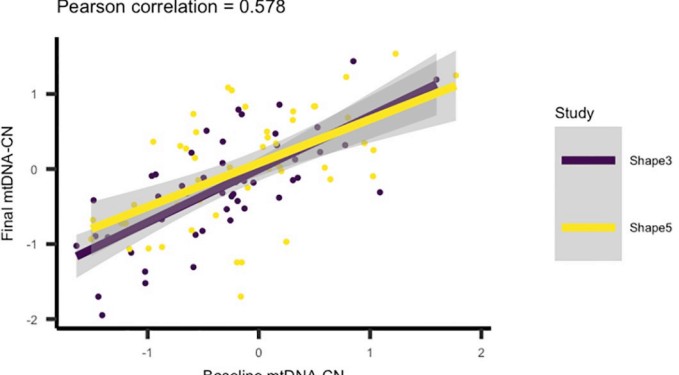

**Fig 2. Strong correlations between baseline and final mtDNA-CN.** Baseline (pre-intervention) and final (post-intervention) mtDNA-CN measurements taken six months apart are well-correlated, with a Pearson correlation of 0.578.

the group of individuals who exercised and found no significant relationship (P = 0.45). However, both analyses were consistent with a positive correlation between mtDNA-CN change and exercise. With our current sample size (N = 105), we had 80% power to detect a 0.234 difference in means between exercisers and non-exercisers.

We also examined associations between mtDNA-CN and VO$_2$max, a measure of cardiorespiratory fitness. A linear mixed model, adjusting for individual as a random effect and age, sex, visit, and study as fixed effects, found no significant associations between mtDNA-CN and VO$_2$max (P = 0.44). There was also no association between 6-month change in mtDNA-CN and 6-month change in VO$_2$max (S7 Fig).

## Evaluating associations between secondary outcomes

In addition to exercise and diet, we were interested in associations between mtDNA-CN and secondary outcomes such as muscle mass, insulin sensitivity, and resting metabolic rate (Table 2). To leverage data from both baseline and final visits, we utilized a linear mixed model, adjusting for age, sex, visit, and individual.

Effect size estimates, standard errors, and p-values from linear mixed models evaluating the relationship between mtDNA-CN and secondary outcomes of interest.

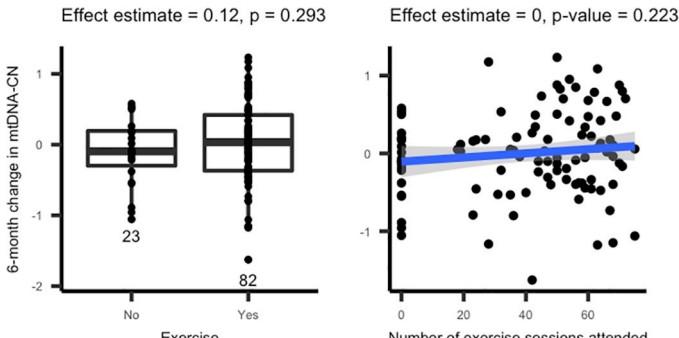

**Fig 3. 6-month change in mtDNA-CN was not associated with exercise.** There was no significant association between exercise intervention and 6-month change in mtDNA-CN.

**Table 2. Associations between secondary outcomes and mtDNA-CN.**

| Secondary outcome | Effect size estimate | Standard error | P-value | FDR-adjusted P-value |
|---|---|---|---|---|
| Muscle mass | -0.25 | 0.38 | 0.51 | 0.63 |
| **Insulin sensitivity** | **0.005** | **0.002** | **0.02** | **0.10** |
| Resting metabolic rate | 0.33 | 7.95 | 0.97 | 0.97 |
| Baseline glycemia | -1.14 | 1.80 | 0.53 | 0.63 |
| **Metabolic syndrome** | **-0.81** | **0.38** | **0.04** | **0.11** |
| HbA1c | 0.09 | 0.06 | 0.13 | 0.26 |

Of these secondary outcomes, insulin sensitivity and metabolic syndrome were both associated with mtDNA-CN prior to multiple-testing correction. Associations between mtDNA-CN and metabolic syndrome have been previously reported, supporting this finding [19]. As individuals with prevalent diabetes are known to have lower mtDNA-CN [20] and type 2 diabetes is a disease primarily characterized by decreased insulin sensitivity [21], we re-examined this association after adjusting insulin sensitivity for diabetes status. The association between mtDNA-CN and insulin sensitivity remained, even after accounting for diabetes status (P = 0.007).

As many of these secondary outcomes are known to be linked with metabolic syndrome, we repeated the analysis, adjusting for metabolic syndrome as a covariate. Results did not significantly change, suggesting that metabolic syndrome does not mediate relationships between mtDNA-CN and these secondary outcomes (S2 Table).

## Discussion

In the SHAPE cohorts, metabolic syndrome and insulin sensitivity were nominally significantly associated with mtDNA-CN, with lower mtDNA-CN associated with metabolic syndrome and lower insulin sensitivity. mtDNA-CN from baseline and final visits were well-correlated, indicating that while mtDNA-CN may change over time, measures taken six months apart are relatively consistent.

We were powered to detect a difference of 0.234 standard deviations for 6-month change in mtDNA-CN between exercising and non-exercising groups and did not observe a significant difference in our dataset. Previous literature has described a significant increase in blood mtDNA-CN after exercise, as well as a significant association between mtDNA-CN and $VO_2$max [22]. However, these methods do not normalize mtDNA content to a nuclear DNA target, normalizing instead to a spike-in standard DNA target. As such, the metrics used in the aforementioned study do not adjust for the number of cells and cell type differentials present in each sample.

After evaluating associations between mtDNA-CN and several secondary outcomes, insulin sensitivity as estimated from the QUICKI score was significantly associated with mtDNA-CN, with higher mtDNA-CN associated with increased insulin sensitivity. Loss of mitochondrial function in elderly subjects has been shown to lead to lipid accumulation and ultimate insulin resistance, corroborating our finding [23].

Since we do not have cell-type composition data, it is difficult to determine whether observed changes in mtDNA-CN are due to changes in mitochondrial content or changes in cell type composition, as aerobic exercise is known to cause decreases in neutrophils [24] and monocyte-platelet aggregates [21], but also causes increases in overall platelet count [25].

mtDNA-CN is known to be confounded by cell-type composition, with increased platelet count leading to higher mtDNA-CN and increased neutrophil count leading to lower mtDNA-CN [26]. As such, these findings must be interpreted with this limitation in mind.

Also, the type of exercise training may affect our ability to detect changes in mitochondrial DNA quantity. Both of the SHAPE cohorts were subjected to both aerobic and resistance exercise training. However, previous studies have shown that ribosomal and mitochondrial biogenesis may be competitive processes, with resistance training favoring ribosomal biogenesis and with aerobic exercise prioritizing mitochondrial biogenesis [27].

An additional constraint to this study is the varying comorbidities between SHAPE3 and SHAPE5. Although both SHAPE cohorts are comprised of obese and overweight individuals, SHAPE3 recruited subjects with type 2 diabetes or having prediabetes, while SHAPE5 individuals were otherwise healthy except for having abdominal obesity. As diabetes is known to cause abnormalities in mitochondrial function [20, 28, 29], this may affect the relationship between exercise and mtDNA-CN in individuals with diabetes. However, with the limited sample size in this study, there did not appear to be an association between exercise and mtDNA-CN, and addition of diabetes status as a covariate did not change results.

In summary, we do not detect a significant change in mtDNA-CN after exercise intervention in these study cohorts, despite marked improvements in fitness and substantial weight loss. After examining secondary outcomes, we uncovered a significant association between mtDNA-CN and insulin sensitivity, likely driven by biological pathways that connect mitochondria, lipid accumulation, and insulin resistance.

## Methods

### Participant recruitment

This study was approved by the Johns Hopkins Medicine IRB under retrospective application IRB0007178. Written informed consent was obtained for all participants, and all DNA samples and associated phenotype data were de-identified prior to analysis. All studies are listed under ClinicalTrials.gov (SHAPE3: NCT00928005, SHAPE5: NCT00990457).

**SHAPE3.** Subjects were overweight or obese (BMI between 26 and 42 kg/m2), sedentary men and women (n = 77), 30–65 years, with prediabetes or diabetes, according to American Diabetes Association criteria (fasting glucose > 126 mg/dl, casual plasma glucose > 200 mg/dl, or 2-hour plasma glucose > 200 mg/dl after a 75-gram oral glucose load). Individuals with uncontrolled diabetes, defined as fasting blood glucose over 300 mg/dl or A1C > 11% were excluded.

**SHAPE5.** Subjects were overweight or obese, sedentary men and women (n = 77), BMI 25–42 kg/m$^2$, 30–65 years, who were otherwise healthy.

### Exercise and diet intervention

**SHAPE3.** Diet intervention for SHAPE3 was a nutritionally balanced, moderately hypocaloric diet with reduced saturated fat consistent with American Diabetes Association guidelines. The diet was adjusted to produce a 600 kcal deficit/day for each individual, using resting metabolic rate calculated from the Mifflin-St Jeor equation [30].

Exercise intervention for SHAPE3 was designed based on guidelines from the American College of Sports Medicine and the American Diabetes Association, consisting of warm-up, 45 minutes of aerobic exercise, several resistance training exercises, and cool-down. Exercise sessions were supervised by exercise physiologists to ensure safety and that the exercises were carried out properly. Individuals assigned to exercise intervention were asked to exercise 3 times a week over a 26-week period.

**SHAPE5.** The low-carbohydrate (CHO) group adhered to the New Atkins for Life diet, consisting initially of 15% CHO, 30% protein, and 55% fat, followed by a gradual shift to 40% CHO, 20% protein, and 40% fat.

The low-fat group followed American Heart Association (AHA) and National Cholesterol Education Program (NCEP) guidelines, following a diet of 30% fat, 50–55% CHO, and 15–20% protein.

All subjects in SHAPE5 participated in 3 times per week supervised exercise training following ACSM guidelines for moderate intensity aerobic and resistance training, consisting of 45 minutes of aerobic exercise and 2 sets of 7 resistance exercises.

## Measurement of study variables

Maximal oxygen uptake (VO2 max ml/kg/min). A Cardinal Health Metabolic/EKG system was used to measure $VO_2$ max. The exercise began at 3 mph, 0% grade, and increased 2.5% grade every 3 minutes. There was continuous EKG and cardiorespiratory monitoring. The 12 lead ECG was recorded at every stage. BP was measured during the last 30 seconds and the Rating of perceived exertion (RPE), using the Borg 6 to 20 scale, was obtained during each stage. An RPE of 18–20 and a respiratory exchange ratio > 1.1 were considered as indicators of maximal effort. The highest observed value of $VO_2$ was recorded as $VO_2$ max.

Muscle mass was measured using Dual Energy X-Ray Absorptiometry (DEXA) with a GE Lunar Prodigy. DEXA lean mass measurements were utilized as a representation for muscle mass. Insulin sensitivity was calculated from fasting glucose and fasting insulin measurements using the quantitative insulin sensitivity check index (QUICKI) formula [31]. Insulin, glucose, and Hb1ac levels were measured from a fasting blood draw. Anthropometry was performed to obtain height and weight measurements with a balance scale and stadiometer. BMI was then calculated using these measurements. Waist and hip measurements were taken using a tape measure. Resting metabolic rate was estimated using the Mifflin-St Jeor equation [30].

Subjects were categorized as having metabolic syndrome if they had $\geq 3$ of the 5 factors: central obesity with a waist circumference of > 40 inches (M) or > 35 inches (F); hyperglycemia, fast glucose $\geq 100$ mg/dl or taking medications; dyslipidemia, triglycerides $\geq 150$ mg/dl or taking medications; dyslipidemia $2^{nd}$, separate criteria, HDL cholesterol $\leq 40$ mg/dl (M) or $\leq 50$ mg/dl (F) or taking medication for both sexes; hypertension, $\geq 130$ mm Hg systolic or $\geq 85$ mm Hg diastolic, or taking medications.

## mtDNA-CN measurement

mtDNA-CN was measured using a monochrome qPCR method [15]. Previous work comparing this assay with mtDNA-CN derived from whole genome sequencing has shown that individuals with polymorphisms in the D-loop primer region have unreliable mtDNA-CN monochrome assay measurements [32]. As such, samples that had deltaCT (difference between nuclear and mitochondrial probe CTs) less than 7 were filtered out due to assumed polymorphisms in the D-loop primer (7 total samples). One outlier individual was removed due to a baseline and a final mtDNA-CN value that was greater than 3 SD from the mean.

## Genetic fingerprinting

Genetic fingerprinting using the Agena iPLEX Pro SampleID Panel was used to identify sample swaps and confirm that baseline and final samples originated from the same individual. Seven samples (two total individuals) were removed due to duplicated sample IDs with non-matching genetic information, as there was no way to match which sample corresponded to the correct individual. Four samples were removed due to poor sample quality, with greater

than 50% missingness on the array. Finally, two samples (one individual) were removed due to fewer than 90% matching calls between baseline and final samples.

## Statistical analyses

All statistical analyses were performed with R version 4.1.1 [33]. Linear mixed models were performed using the lme4 package and plots were created with ggplot2 [34, 35].

## Supporting information

**S1 Fig. Exercise is associated with an increase in VO2max.** The number of exercise sessions attended is significantly associated with a 6-month increase in $VO_2$max.
(TIF)

**S2 Fig. Exercise is associated with a decrease in BMI.** There was a significant association between a greater number of exercise sessions attended and decreased BMI over the 6-month intervention period.
(TIF)

**S3 Fig. mtDNA-CN is associated with known covariates in both cohorts.** Associations between mtDNA-CN and age and sex are in the expected directions when stratifying by SHAPE study. SHAPE3 on the left, SHAPE5 on the right.
(TIF)

**S4 Fig. mtDNA-CN is correlated with study retention.** Individuals who dropped out of the study had significantly higher mtDNA-CN.
(TIF)

**S5 Fig. Correlation with study dropout appears in both cohorts.** Individuals who dropped out of the study have significantly higher mtDNA-CN than those who were retained (On left, SHAPE3, on right, SHAPE5).
(TIF)

**S6 Fig. Baseline mtDNA-CN is associated with a change in mtDNA-CN in the direction of the mean.** mtDNA-CN measured at baseline is associated with a change in mtDNA-CN in the direction of the mean. Purple points denote individuals with final mtDNA-CN measurements closer to the mean than their baseline measurements, while yellow is vice versa. Significantly more individuals move towards the mean (chi-squared p = 0.004). For the purple samples, absolute magnitude of baseline mtDNA-CN is positively correlated with the absolute value of 6-month change, however, this association is not significant (P = 0.51).
(TIF)

**S7 Fig. $VO_2$ max and change in mtDNA-CN are not associated.** There was no significant association between 6-month change in $VO_2$ max and 6-month change in mtDNA-CN.
(TIF)

**S1 Table. Odds ratios and p-values for associations between explanatory variables and study completion.** There was no significant relationship between study completion and variables of interest, save for age. However, the directionality of the association between age and study completion does not explain the relationship between higher mtDNA-CN and higher study dropout.
(XLSX)

**S2 Table. Adjusting for metabolic syndrome does not change results.** Effect size estimates for secondary outcomes after including metabolic syndrome status as a covariate. Results generally stay the same, indicating that metabolic syndrome is not mediating the effects of mtDNA-CN on these outcomes.
(XLSX)

## Acknowledgments

The authors gratefully acknowledge use of the facilities at the Joint High Performance Computing Exchange (JHPCE) in the Department of Biostatistics, Johns Hopkins Bloomberg School of Public Health that have contributed to the results reported within this paper.

## Author Contributions

**Conceptualization:** Kerry J. Stewart, Dan E. Arking.

**Data curation:** Stephanie Y. Yang, Caleb S. Mirabal, Charles E. Newcomb, Kerry J. Stewart.

**Formal analysis:** Stephanie Y. Yang.

**Funding acquisition:** Kerry J. Stewart, Dan E. Arking.

**Investigation:** Stephanie Y. Yang, Dan E. Arking.

**Methodology:** Stephanie Y. Yang, Caleb S. Mirabal, Charles E. Newcomb, Kerry J. Stewart, Dan E. Arking.

**Project administration:** Charles E. Newcomb, Kerry J. Stewart.

**Resources:** Kerry J. Stewart, Dan E. Arking.

**Supervision:** Dan E. Arking.

**Visualization:** Stephanie Y. Yang.

**Writing – original draft:** Stephanie Y. Yang.

**Writing – review & editing:** Kerry J. Stewart, Dan E. Arking.

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
