## [Decision Letter · Decision Letter 0]

12 May 2022

PONE-D-21-37204Mitochondrial DNA copy number and insulin sensitivity: Insights from the Sugar, Hypertension, and Physical Exercise StudiesPLOS ONE

Dear Dr. Arking,

Thank you for submitting your manuscript to PLOS ONE. After careful consideration, we feel that it has merit but does not fully meet PLOS ONE’s publication criteria as it currently stands. Therefore, we invite you to submit a revised version of the manuscript that addresses the points raised during the review process.

We look forward to receiving your revised manuscript.

Kind regards,

Victoria J. Vieira-Potter

Academic Editor

PLOS ONE

**Journal requirements:**

“This work was supported by grants R01HL13573 and R01HL144569. SHAPE projects were supported by grants R01DK062368 and UL1RR025005.”

**Additional Editor Comments:**

Due to this unfortunate circumstance of of only one peer review, I am making the decision "major revision". Please refer to the reviewer's comments. The decision was reject, but I do not feel comfortable rejecting the paper outright based on only one review. Hopefully these comments will be helpful as you revise your work either to resubmit to PLOS one or elsewhere.

Best Regards,

VVP

Reviewers' comments:

Reviewer's Responses to Questions

**Comments to the Author**

1. Is the manuscript technically sound, and do the data support the conclusions?

Reviewer #1: Partly

2. Has the statistical analysis been performed appropriately and rigorously? 

Reviewer #1: I Don't Know

3. Have the authors made all data underlying the findings in their manuscript fully available?

Reviewer #1: Yes

4. Is the manuscript presented in an intelligible fashion and written in standard English?

Reviewer #1: Yes

5. Review Comments to the Author

Reviewer #1: This study capitalized on a rich human dataset, including V02 max data and pre and post exercise-intervention data including several metabolic variables on obese individuals. This study correlates these data with circulating levels of mtDNA-CN, an emerging indicator of metabolic health. Although interesting, the new data provided (above what has already been published in the SHAPE interventions) do not seem to warrant publication on their own. Most of the data are not significant, yet some are confirmatory, which is somewhat beneficial to the field. It is particularly interesting that there are age and sex differences. The data do not seem to support strongly a relationship between exercise training and mtDNA-CN, but perhaps the lack of relationship was due to the population mostly being obese and not highly fit. The vast majority of data are presented as supplementary figures, indicating that most of the content is not deemed of sufficient value to most readers (by the authors). There are some interesting findings here. It would be of value if the authors did have skeletal muscle biopsy data to relate to the mtDNA-CN data. This would highlight the potential clinical value that mtDNA-CN may have, leading future studies to use blood mtDNA-CN as an indicator of skeletal muscle quality/mitochondrial activity. The finding that baseline values predicted dropout is peculiar and interesting on its own. If the authors could dig deeper into the data to find a potential explanation for this finding, that could be a central novel element of the paper. Unfortunately, as it currently reads, the paper lacks a major novel finding. It appears that the major finding is that exercise training does not affect mtDNA-CN in obese individuals. Maybe this is an important enough finding to warrant publication, but I just was not convinced of this, especially given the study limitations. I offer my suggestions below and hope this comments are helpful to the authors.

Introduction:

Line 50, Do you mean “positively” correlated? Are there ever instances where mtDNA-CN are elevated in conditions where mitochondrial are dysfunction, such as with dysfunction in mitophagy pathways leading to greater numbers of dysfunctional mitochondria?

Line 52, Please describe “buffy coat” for readers unfamiliar with blood analysis

Line 58, has shown should be have shown

Line 59, for should be of?

More detailed description of “exercise training” is required. Type of exercise? Subject population? Duration/chronicity/intensity?

Results:

The sex differences are interesting and important. How does age affect the sex difference? Have any studies addressed how estrogen or ot her sex hormones may affect mtDNA-CN?

The “effect estimates” that are described – are there p values associated with these graphs (Supp Fig 3)?

The dropout/baseline mtDNA-CN values: a more detailed explanation of what potential covariates were ruled out to explain this relationship would be helpful. (eg, age? BMI? Some sociological variable?)

6. PLOS authors have the option to publish the peer review history of their article (what does this mean?). If published, this will include your full peer review and any attached files.

Reviewer #1: No

---

## [Author Response · Author response to Decision Letter 0]

16 Jun 2022

We thank the reviewer for their comments, and have responded to them below.

Reviewer #1: This study capitalized on a rich human dataset, including V02 max data and pre and post exercise-intervention data including several metabolic variables on obese individuals. This study correlates these data with circulating levels of mtDNA-CN, an emerging indicator of metabolic health. Although interesting, the new data provided (above what has already been published in the SHAPE interventions) do not seem to warrant publication on their own. Most of the data are not significant, yet some are confirmatory, which is somewhat beneficial to the field. It is particularly interesting that there are age and sex differences. The data do not seem to support strongly a relationship between exercise training and mtDNA-CN, but perhaps the lack of relationship was due to the population mostly being obese and not highly fit. The vast majority of data are presented as supplementary figures, indicating that most of the content is not deemed of sufficient value to most readers (by the authors). There are some interesting findings here. It would be of value if the authors did have skeletal muscle biopsy data to relate to the mtDNA-CN data. This would highlight the potential clinical value that mtDNA-CN may have, leading future studies to use blood mtDNA-CN as an indicator of skeletal muscle quality/mitochondrial activity. The finding that baseline values predicted dropout is peculiar and interesting on its own. If the authors could dig deeper into the data to find a potential explanation for this finding, that could be a central novel element of the paper. Unfortunately, as it currently reads, the paper lacks a major novel finding. It appears that the major finding is that exercise training does not affect mtDNA-CN in obese individuals. Maybe this is an important enough finding to warrant publication, but I just was not convinced of this, especially given the study limitations. I offer my suggestions below and hope this comments are helpful to the authors.

We thank the reviewer for their insightful comments and suggestions. While we agree with the reviewer’s conclusions that the current data does not support a relationship between exercise training and mtDNA-CN, we would like to note the criteria for publication in PLoS One (taken directly from the website) are as follows: 

1. The study presents the results of original research. 

2. Results reported have not been published elsewhere.

3. Experiments, statistics, and other analyses are performed to a high technical standard and are described in sufficient detail.

4. Conclusions are presented in an appropriate fashion and are supported by the data.

5. The article is presented in an intelligible fashion and is written in standard English.

6. The research meets all applicable standards for the ethics of experimentation and research integrity.

We believe that the current study and all analyses were performed with scientific rigor, and that all conclusions in the paper are supported by the data. Furthermore, the manuscript is written in standard English, has not been published elsewhere, and describes the results of original research. Research was conducted in accordance with IRB protocols (IRB00071780), and all clinical studies are listed on ClinicalTrials.gov (SHAPE3: NCT00928005, SHAPE5: NCT00990457). 

Given this, we believe that this manuscript meets all the criteria for publication in PLoS One. Though negative results are not as exciting or flashy, they are still important and should be disseminated to the rest of the research community. 

Introduction:

Line 50, Do you mean “positively” correlated? Are there ever instances where mtDNA-CN are elevated in conditions where mitochondrial are dysfunction, such as with dysfunction in mitophagy pathways leading to greater numbers of dysfunctional mitochondria?

We thank the reviewer for this suggestion and have clarified and adjusted the text on line 50 to read as follows:

Higher mtDNA-CN levels are positively associated with mitochondrial membrane potential, respiratory enzyme function, and energy reserves [1,2]… 

There have been instances where elevated mtDNA-CN is associated with cancer, however, we have not seen literature where elevated mtDNA-CN is associated with mitochondrial dysfunction. 

Line 52, Please describe “buffy coat” for readers unfamiliar with blood analysis

We have adjusted the text on lines 52-53 to define buffy coat:

Lower mtDNA-CN in buffy coat, the fraction of blood that contains leukocytes and platelets, has been associated with frailty, often characterized by decreased muscle tone [3].

Line 58, has shown should be have shown

We disagree with this suggestion, as we believe version 1 of the sentence is more grammatically sound than version 2.

[1] Indeed, persistent exercise training has been shown to increase mitochondrial function and mitochondrial volume in skeletal muscle [4,5].

[2] Indeed, persistent exercise training have shown to increase mitochondrial function and mitochondrial volume in skeletal muscle [4,5].

Line 59, for should be of?

We have corrected this sentence.

More detailed description of “exercise training” is required. Type of exercise? Subject population? Duration/chronicity/intensity?

Later on in the Methods section of the paper, we rigorously describe the type of exercise (lines 271-291), the subject populations (lines 262-269), and the duration/chronicity/intensity (lines 271-291). We agree with the reviewer that increased detail in the introduction would be useful, and have included some more details in that section (lines 61-64):

To answer this question, we used two randomized controlled exercise intervention cohorts. Participants were aged 30-65 years, and performed aerobic and resistance exercise training 3 times a week for a duration of 6 months. 

We have included the more detailed methods sections below for ease of reference:

SHAPE3

Subjects were overweight or obese (BMI between 26 and 42 kg/m2), sedentary men and women (n=77), 30-65 years, with prediabetes or diabetes, according to American Diabetes Association criteria (fasting glucose > 126 mg/dl, casual plasma glucose > 200 mg/dl, or 2-hour plasma glucose > 200 mg/dl after a 75-gram oral glucose load). Individuals with uncontrolled diabetes, defined as fasting blood glucose over 300 mg/dl or A1C > 11% were excluded.

SHAPE5

Subjects were overweight or obese, sedentary men and women (n=77), BMI 25-42 kg/m2, 30-65 years, who were otherwise healthy.

Exercise and Diet Intervention

SHAPE3 

Diet intervention for SHAPE3 was a nutritionally balanced, moderately hypocaloric diet with reduced saturated fat consistent with American Diabetes Association guidelines. The diet was adjusted to produce a 600 kcal deficit/day for each individual, using resting metabolic rate calculated from the Mifflin-St Jeor equation [6].

Exercise intervention for SHAPE3 was designed based on guidelines from the American College of Sports Medicine and the American Diabetes Association, consisting of warm-up, 45 minutes of aerobic exercise, several resistance training exercises, and cool-down. Exercise sessions were supervised by exercise physiologists to ensure safety and that the exercises were carried out properly. Individuals assigned to exercise intervention were asked to exercise 3 times a week over a 26-week period. 

SHAPE5

The low-carbohydrate (CHO) group adhered to the New Atkins for Life diet, consisting initially of 15% CHO, 30% protein, and 55% fat, followed by a gradual shift to 40% CHO, 20% protein, and 40% fat.

The low-fat group followed American Heart Association (AHA) and National Cholesterol Education Program (NCEP) guidelines, following a diet of 30% fat, 50-55% CHO, and 15-20% protein.

All subjects in SHAPE5 participated in 3 times per week supervised exercise training following ACSM guidelines for moderate intensity aerobic and resistance training, consisting of 45 minutes of aerobic exercise and 2 sets of 7 resistance exercises.

Results:

The sex differences are interesting and important. How does age affect the sex difference? Have any studies addressed how estrogen or other sex hormones may affect mtDNA-CN?

It has been well-established that mtDNA-CN varies between sexes[7,8]. While we agree with the reviewer that these questions are interesting and important, we believe that they fall beyond the scope of this manuscript, as the goal of this study was to examine the effect of exercise on mtDNA-CN. 

The “effect estimates” that are described – are there p values associated with these graphs (Supp Fig 3)?

We thank the reviewer for this observation, and agree that this information should be included. We have revised the figure to include effect size estimates and p-values. As a spline does not significantly improve the model for mtDNA-CN and age, we have used a simple linear regression for the plots that are stratified by study. 

The dropout/baseline mtDNA-CN values: a more detailed explanation of what potential covariates were ruled out to explain this relationship would be helpful. (eg, age? BMI? Some sociological variable?)

To clarify covariates that were examined, we have added details on the potential underlying covariates that we evaluated for correlation with dropout (lines 118-126), along with a table (Supplemental Table 1). 

To understand what could be driving the relationship between baseline mtDNA-CN and study completion, we examined associations between study completion and several other variables. However, none of these potential explanatory variables was significantly associated with study completion. We note that increased age was nominally significantly associated with dropout, however, since increased age is associated with lower mtDNA-CN, this would not explain the observed relationship between higher mtDNA-CN and increased rates of study dropout (Supplemental Table 1). After these analyses, we were unable to account for the variation in dropout explained by baseline mtDNA-CN, and currently do not have a biological explanation for this finding.

Variable Name Odds Ratio p-value

Age 1.055 0.012

BMI 1.018 0.679

Insulin level 1.004 0.734

Sleep score 1.007 0.760

C-reactive protein 1.014 0.734

Supplemental Table 1. Odds ratios and p-values for associations between explanatory variables and study completion. There was no significant relationship between study completion and variables of interest, save for age. However, the directionality of the association between age and study completion does not explain the relationship between higher mtDNA-CN and higher study dropout.

References

1. Guha M, Avadhani NG. Mitochondrial retrograde signaling at the crossroads of tumor bioenergetics, genetics and epigenetics. Mitochondrion. 2013;13: 577–591. doi:10.1016/j.mito.2013.08.007

2. Jeng J-Y, Yeh T-S, Lee J-W, Lin S-H, Fong T-H, Hsieh R-H. Maintenance of mitochondrial DNA copy number and expression are essential for preservation of mitochondrial function and cell growth. J Cell Biochem. 2008;103: 347–357. doi:10.1002/jcb.21625

3. Ashar FN, Moes A, Moore AZ, Grove ML, Chaves PHM, Coresh J, et al. Association of Mitochondrial DNA levels with Frailty and All-Cause Mortality. J Mol Med Berl Ger. 2015;93: 177–186. doi:10.1007/s00109-014-1233-3

4. Jacobs RA, Lundby C. Mitochondria express enhanced quality as well as quantity in association with aerobic fitness across recreationally active individuals up to elite athletes. J Appl Physiol. 2013;114: 344–350. doi:10.1152/japplphysiol.01081.2012

5. Menshikova EV, Ritov VB, Fairfull L, Ferrell RE, Kelley DE, Goodpaster BH. Effects of Exercise on Mitochondrial Content and Function in Aging Human Skeletal Muscle. J Gerontol A Biol Sci Med Sci. 2006;61: 534–540. 

6. Mifflin MD, St Jeor ST, Hill LA, Scott BJ, Daugherty SA, Koh YO. A new predictive equation for resting energy expenditure in healthy individuals. Am J Clin Nutr. 1990;51: 241–247. doi:10.1093/ajcn/51.2.241

7. Tin A, Grams ME, Ashar FN, Lane JA, Rosenberg AZ, Grove ML, et al. Association between Mitochondrial DNA Copy Number in Peripheral Blood and Incident CKD in the Atherosclerosis Risk in Communities Study. J Am Soc Nephrol JASN. 2016;27: 2467–2473. doi:10.1681/ASN.2015060661

8. Knez J, Winckelmans E, Plusquin M, Thijs L, Cauwenberghs N, Gu Y, et al. Correlates of Peripheral Blood Mitochondrial DNA Content in a General Population. Am J Epidemiol. 2016;183: 138–146. doi:10.1093/aje/kwv175

---

## [Editor Report · Decision Letter 1]

22 Jun 2022

Mitochondrial DNA copy number, metabolic syndrome, and insulin sensitivity: Insights from the Sugar, Hypertension, and Physical Exercise Studies

PONE-D-21-37204R1

Dear Dr. Arking,

We’re pleased to inform you that your manuscript has been judged scientifically suitable for publication and will be formally accepted for publication once it meets all outstanding technical requirements.

Kind regards,

Hans-Peter Kubis, PD. Dr. rer. nat.

Academic Editor

PLOS ONE

Additional Editor Comments (optional):

Dear Dr Arking,

Thank you for the resubmission of your manuscript. The manuscript was reviewed in the current form and is now acceptable for publication in PLOS ONE. We thank you for submitting your work to PLOS ONE and hope to see more manuscripts being sent to us in the future. Many thanks for considering PLOS ONE and good luck for your future research.

---

## [Editor Report · Acceptance letter]

8 Jul 2022

PONE-D-21-37204R1 

Mitochondrial DNA copy number, metabolic syndrome, and insulin sensitivity: Insights from the Sugar, Hypertension, and Physical Exercise Studies  

Dear Dr. Arking:

I'm pleased to inform you that your manuscript has been deemed suitable for publication in PLOS ONE. Congratulations! Your manuscript is now with our production department. 

Kind regards, 

on behalf of

Dr. Hans-Peter Kubis 

Academic Editor

PLOS ONE